# Determinants of utilization of antenatal and delivery care at the community level in rural Bangladesh

Jesmin Pervin[1,2]*, Mahima Venkateswaran[2,3], U. Tin Nu[1], Monjur Rahman[1], Brian F. O'Donnell[3], Ingrid K. Friberg[3,4], Anisur Rahman[1], J. Frederik Frøen[2,3]

1 International Centre for Diarrhoeal Disease Research, Dhaka, Bangladesh, 2 University of Bergen, Bergen, Norway, 3 Norwegian Institute of Public Health, Oslo, Norway, 4 Tacoma-Pierce County Health Department, Tacoma, WA, United States of America

* jpervin@icddrb.org

**Data Availability Statement:** All relevant data are available within the manuscript and its Supporting Information files.

## Abstract

### Background

Timely utilization of antenatal care and delivery services supports the health of mothers and babies. Few studies exist on the utilization and determinants of timely ANC and use of different types of health facilities at the community level in Bangladesh. This study aims to assess the utilization, timeliness of, and socio-demographic determinants of antenatal and delivery care services in two sub-districts in Bangladesh.

### Methods

This cross-sectional study used data collected through a structured questionnaire in the eRegMat cluster-randomized controlled trial, which enrolled pregnant women between October 2018-June 2020. We undertook univariate and multivariate logistic regression analysis to determine the associations of socio-demographic variables with timely first ANC, four timely ANC visits, and facility delivery. We considered the associations in the multivariate logistic regression as statistically significant if the p-value was found to be <0.05. Results are presented as adjusted odds ratios (AOR) with 95% confidence intervals (CI).

### Results

Data were available on 3293 pregnant women. Attendance at a timely first antenatal care visit was 59%. Uptake of four timely antenatal care visits was 4.2%. About three-fourths of the women delivered in a health facility. Women from all socio-economic groups gradually shifted from using public health facilities to private hospitals as the pregnancy advanced. Timely first antenatal care visit was associated with: women over 30 years of age (AOR: 1.52, 95% CI: 1.05–2.19); nulliparity (AOR: 1.30, 95% CI: 1.04–1.62); husbands with >10 years of education (AOR: 1.40, 95% CI: 1.09–1.81) and being in the highest wealth quintile (AOR: 1.49, 95% CI: 1.18–1.89). Facility deliveries were associated with woman's age; parity; education; the husband's education, and wealth index. None of the available socio-demographic factors were associated with four timely antenatal care visits.

**Funding:** This research is part of the eRegistries Bangladesh project funded by the Norwegian Research Council (grant agreement number 248073/H10; title: Strengthening the extension of Reproductive, Maternal, Newborn, and Child Health services in Bangladesh with an electronic health registry: A cluster randomized controlled trial), and the Centre for Intervention Science in Maternal and Child Health (CISMAC), Center for International Health, University of Bergen (project number: 223269). icddr,b is also grateful to the Governments of Bangladesh, Canada, Sweden and the UK for providing core/unrestricted support. The funders had no role in the design of the study and collection, analysis, and interpretation of data.

**Competing interests:** The authors have declared that no competing interests exist.

**Abbreviations:** ANC, Antenatal care; AOR, Adjusted Odds Ratio; eRegMat, electronic registry Matlab; DHIS2, District Health Information System 2; CC, Community Clinic; UH&FWC, Union Health and Family Welfare Centre; MOHFW, Ministry of Health and Family Welfare; DGFP, Directorate General of Family Planning; DGHS, Directorate General of Health Services; FWV, Family Welfare Visitor; FWA, Family Welfare Assistant; CHCP, Community Health Care Provider; HA, Health Assistant; NGO, Non-government organization; icddr,b, International Centre for Diarrheal Disease, Bangladesh.

## Conclusions

The study observed socio-demographic inequalities associated with increased utilization of timely first antenatal care visit and facility delivery. The pregnant women, irrespective of wealth shifted from public to private facilities for their antenatal care visits and delivery. To increase the health service utilization and promote good health, maternal health care programs should pay particular attention to young, multiparous women, of low socio-economic status, or with poorly educated husbands.

## Clinical trial registration

ISRCTN69491836; https://www.isrctn.com/. Registered on December 06, 2018. Retrospectively registered.

## Introduction

Timely utilization of antenatal care (ANC) and delivery services is important to support maternal health and allow for the best possible start to babies' lives. Most pregnancy complications leading to severe morbidity or mortality can be prevented or treated if women use healthcare services provided by skilled providers with the necessary resources according to recommended clinical guidelines [1]. Use of health services during pregnancy allows for recognition of warning signs of complications, referral of mothers to emergency care, and management of severe complications during pregnancy and childbirth. Despite progress in the use of maternal health services in some low- and middle-income countries, further increase is needed to improve maternal and neonatal health [2].

ANC utilization is crucial for timely identification, prevention, and management of factors influencing pregnancy outcomes [3–5]. While significant progress has been made worldwide in the use of health facilities during childbirth (77%) [6], only about 64% of women receive four or more ANC visits [7,8]. In South Asia, the uptake of four or more ANC visits is about 49% [9]. Late initiation of ANC is one reason for underutilization of care that hinders women and their family's introduction to the formal health system, leading to an increased risk of adverse pregnancy outcomes [5,7].

Bangladesh has decreased maternal and neonatal mortality remarkably from 1990 to 2018. Maternal mortality has been reduced from 570 to 196 per 100,000 live births during that period [10]. For the same period, neonatal mortality has been reduced from 59 to 30 per 1000 live births [11,12]. However, the levels of maternal and neonatal mortality have remained stagnant since 2015, the end of the Millennium Development Goals (MDG) era. Sustainable Development Goal 3 aims for the end of all preventable deaths and sets the target to reduce the maternal mortality ratio to less than 70 per 100,000 live births and neonatal mortality to at least as low as 12 per 1,000 live births by 2030 [13]. To achieve these targets, better health care utilization in both pregnancy and delivery is essential.

In Bangladesh, the use of health facilities for ANC and childbirth has been steadily increasing over the years. About 47% of women receive four or more ANC services. However, only 37% of women receive their first ANC before 16 weeks and 8% of pregnant women still do not receive any ANC [14]. About half of all deliveries occur in health facilities; most of them are privately-owned [14]. The use of public and non-governmental health facilities for childbirth has also increased, but to a lesser extent. There are considerable inequalities in the utilization

of maternal health services in the different sub-districts of Bangladesh [14]. A recent cross-sectional study conducted in a sub-district of the Noakhali district found lower utilization of ANC services (34.6%) and facility delivery (5.3%) compared to the reported national statistics [15].

The evaluation of utilization and determinants of timely ANC is essential to improve the maternal and neonatal health outcomes. Studies in Bangladesh have reported many factors associated with the use of ANC services, such as age, religion, parity, having a living child, educational attainment of women, place of residence, household wealth status, decision-making power, complications during the current pregnancy or a previous pregnancy, the husband's education, and access to mass media [16–18]. Similar factors have been identified as determinants of institutional delivery [15,19,20].

Nevertheless, very few studies exist on either the utilization or determinants of timely ANC in Bangladesh, and there is a specific lack of community-level information on the timely use of different types of services for ANC and delivery care. To effectively improve women's care-seeking during pregnancy and childbirth information on the utilization patterns of public, private, and non-government organization (NGO) health facilities, based on clients' socio-economic and demographic differences is essential. The present study aims to assess the utilization, timeliness of, and socio-demographic determinants of antenatal and childbirth care services in two sub-districts in Bangladesh.

## Methods

### Study design

This cross-sectional study used data collected as part of a cluster-randomized controlled trial, eRegMat, conducted in two sub-districts, Matlab South and Matlab North under Chandpur district of Bangladesh (trial registration: ISRCTN69491836) [21]. Women with pregnancies identified and registered from October 2018 to June 2020 were enrolled in the eRegMat trial.

### Study setting

The estimated population of Matlab South and Matlab North sub-districts is approximately 200,000 and 300,000, respectively [22]. The Government of Bangladesh recommends four focused ANC visits for all low-risk pregnancies based on the 2002 WHO recommendations: first ANC visit (within $16+^6$ weeks of gestation), second ANC visit ($24+^0$ to $28+^6$ weeks of gestation), third ANC visit (at $32+^6$ weeks of gestation) and fourth ANC visit (at $36+^6$ weeks of gestation) [8]. The Ministry of Health and Family Welfare (MOHFW) provides maternal and child health services through two divisions: the Directorate General of Family Planning (DGFP) and the Directorate General of Health Services (DGHS). In each union Family Welfare Visitors (FWV) and Community Health Care Providers (CHCP) provide care at Health & Family Welfare Centres (UH&FWC) and Community Clinics (CC), respectively. Family Welfare Assistants (FWA) and Health Assistants (HA) provide community outreach services and are the first contacts for the population at the household level. Matlab Health and Research Centre, run by the International Centre for Diarrhoeal Disease Research, Bangladesh (icddr, b), and a few non-government organizations (NGOs) also provide maternal and child health care. Since 2018, a digital maternal and child health registry (eRegistry) has been implemented in two sub-districts for use by both CHCP and FWV in health facilities and by HA and FWA for community-level services. The eRegistry is designed so both facility-based and community-based health workers can access an individual's client record and input clinical data. All pregnancies are supposed to be registered in the eRegistry so as to create comprehensive client records. In total, 72 health facilities were included in the eRegistry roll-out, and a cluster-

randomized controlled trial (eRegMat) was embedded in the implementation. In health facilities assigned to the intervention group (n = 30), three digital health interventions were implemented in addition to the digital longitudinal tracking–clinical decision support, feedback dashboards for health workers, and targeted client communication via SMS to pregnant women. The control group (n = 29) facilities used an eRegistry without additional digital health interventions. Pregnancy registrations in a randomized health facility were automatically allocated to their respective intervention or control group, while community registrations received a trial allocation based on the woman's choice of health facility for ANC. Health facilities that were not included in the trial received the eRegistry without digital health interventions to maintain continuity of data and care in the health system and were classified as non-randomized (n = 13).

For this analysis, we included data on all women in the control and non-randomized groups of the eRegMat trial, as well as women registered in the eRegistry by community-level health workers without trial allocation. Women randomized to the intervention group were excluded.

## Data availability

Data were collected from women within eight to fourteen days after childbirth. For a few cases, the data collection period was extended up to nine months, either because women were not available within eight to fourteen days after childbirth or due to the COVID-19 pandemic lockdown measures. Written consent was obtained for the postpartum survey during pregnancy registration in the eRegistry. A structured questionnaire was prepared for the survey, and the questionnaire was pretested before data collection began. Data were collected on utilization of ANC and delivery services, birth outcomes, and respondents' socio-economic characteristics. Twenty female data collectors with experience in collecting data in household surveys from the same community were recruited and trained for data collection. Two data collectors were appointed to call pregnant women every other week after 28 weeks of gestation and through 35 weeks of gestation and then once a week until their delivery. A monitoring dashboard was developed to identify pregnant women for phone calls based on the gestational age in the eRegistry and collect their pregnancy outcome information. From the monitoring dashboard, one field research assistant produced daily lists of enrolled women who had a pregnancy outcome and distributed those lists to the data collectors. Data collectors then visited the women to conduct the interview after childbirth. After data collection, the survey questionnaires were checked for completeness and discrepancies by the data collectors' supervisors. Data were entered into a web-based electronic form by assigned data entry staffs.

## Outcome variables

The outcome variables and definitions were as follows: 1) timely first ANC visit: a visit within $17+^6$ weeks of gestation; 2) four timely ANC visits: ANC visits at or before $17+^6$ weeks, $24+^0$ to $28+^6$ weeks, $31+^0$ to $33+^6$ weeks and $35+^0$ to $37+^6$ weeks of gestation according to the national ANC schedule [23]; and 3) facility delivery: a delivery in any health facility, including public, private, and those run by icddr,b and NGOs. In order not to underestimate an acceptable timeliness in use of ANC services, and allow for maternal flexibility to attend ANC, we expanded the specific weeks indicated by the guideline by an additional one-week range, except when the guideline already recommended a range of weeks (for example, $24+^0$ to $28+^6$ weeks), where we kept the original range.

We also analyzed the associations between sociodemographic determinants and a Skilled Birth Attendant (SBA) at delivery. We defined a delivery as conducted by a SBA if it was

conducted by a qualified doctor, nurse, midwife, paramedic, FWV or community skilled birth attendant (CSBA) as described in the Bangladesh Health and Demographic Survey [14].

## Explanatory variables

The predictor variables considered in the analysis were the women's age, parity, education, their husband's education, and the household wealth index. Parity was defined as the number of times that a woman had given birth to a baby with a gestational age of 28 weeks or more, regardless of whether it was a live birth or stillbirth. Educational status was recorded by the number of completed years of schooling. The household wealth index was calculated by generating scores through principal-components analysis based on household assets of ownership of a number of consumer items (freezer, television, and others), household livestock, dwelling characteristics (wall and roof material), type of drinking water, toilet facilities, type of fuel mainly used for cooking, and source of electricity. These scores were then indexed into quintiles, where one represented the poorest and five the richest [24]. The last menstrual period date was determined by recall during the interview at the household visit for consistency. Gestational age at each ANC visit was measured by subtracting the LMP date from the ANC visit date and expressed in weeks and days.

## Sample size

The total available sample size was 3293 pregnancies. Power calculations were made on the study outcomes. According to the Bangladesh health and demographic survey report and a recent study conducted in Bangladesh we could expect 50% of women to have a facility delivery, 37% to attend a first timely ANC, and 1% to attend four timely ANC visits during the study period [14,25]. We calculated the power to estimate the prevalence for the facility delivery, timely first ANC, and four timely ANC with 3% error, and an α of 0.05 and the power was estimated to be > 90%. We also performed power calculation for logistic regression analysis and the power was found to be > 80%. We used the "power oneproportion" and "powerlog" commands in Stata version 16 for the power calculations respectively [26].

## Data analysis

We categorized maternal age into <20; 20–30; and >30 years, parity into 0; 1; and ≥2, and education into 0–5; 6–10; and > 10 years of schooling. Quintiles of asset scores were used to categorize socio-economic status. A first ANC visit was considered to be timely if care was received within week 17+[6] of gestational age. All other first ANC visits were considered not timely. Similarly, the other three routine ANC visits were defined timely if they occurred within the time periods defined above. Women were considered to have completed four timely ANC visits if one timely ANC visit was recorded within each of the four recommended time periods. Place of delivery was categorized into home or facility delivery. We used descriptive statistics to present women's socio-demographic characteristics and the utilization of antenatal and delivery care using percentage distribution, mean, and median. The associations between the independent and explanatory variables were tested by Pearson's chi-square ($\chi2$) tests. We assessed all the variables presented in the study in a directed acyclic graph (DAG) approach and found all to be potentially independent confounders [27]. Multicollinearity between the explanatory variables was checked using correlation coefficient and variance inflation factor (VIF). We used the cut off value for the correlation coefficient at ≥ 0.80 and ≥ 5 for VIF. The correlation coefficient between the woman's education and her husband's education was >0.8, but all the values of VIF were <3.

We evaluated the associations between socio-demographic variables and timely first ANC, four timely ANC visits, and facility delivery by univariate and multivariate logistic regressions. All the socio-demographic variables related to the outcomes of interest were included in the multivariate model to adjust for potential confounding, as removing either the woman's education or her husband's education due to potential multicollinearity did not change results substantially. The associations in the multivariate logistic regression were considered statistically significant if the p-value was <0.05. The results of both the univariate and multivariate logistic regression analyses are presented by odds ratios (OR) with 95% confidence intervals (CI). We performed Hosmer and Lemeshow's goodness-of-fit test to identify that our model had a good fit with a p-value>0.05. All statistical analyses were done in Stata version 16 (Stata-Corp, College Station, TX, USA) [26].

### Ethics approval and consent to participate

The study was approved by the Research and Ethical Review Committees of the International Centre for Diarrhoeal Disease Research, Bangladesh, and the Regional Ethical Committee in Norway, Southeast region. All participants received an explanation of the purpose of the study and gave written informed consent for participation in the study.

## Results

### Socio-demographic characteristics

We included a total of 3293 women in the analysis. Of all the women, 84% were interviewed within 8–14 days of childbirth. The characteristics of the women are shown in Table 1. The participants' mean age was 24 years (SD ±4.5), and 18% were under the age of 20 years. The median parity was 2, and 40% of participants were nulliparous, while one-fourth of the participants had 2 or more children. The median number of years of school attendance for study participants and their husbands was 9 and 8 years, respectively.

### ANC utilization

Almost all participants (98%) received ANC at least once, while 91% of women received ANC twice, 74% received ANC thrice and a half (52%) received ANC four or more times. The mean gestational age (GA) at first, second, third and fourth ANC was 17.6 weeks (SD ±6.9), 24.3 weeks (SD ±6.8), 28.4 weeks (SD± 5.9) and 31.1 weeks (±4.9) respectively. More than half of the participants (59%) attended a timely first ANC and 62%, 42% and 31% of women received timely 2nd, 3rd and 4th ANC, respectively. Overall, 94% received timely ANC once, 68% twice and 22% thrice. However, only 4.2% of women attended ANC timely four times in line with the recommended ANC schedule.

Among the women who attended at least one ANC (Fig 1), about 44% attended their first ANC in public health facilities, whereas 40% of women visited private facilities. On the other hand, 26% of women received their 4th ANC in private facilities, and only 17% in public facilities.

For ANC visits after the first visit (Fig 2), the public health sector gradually lost more women compared to the private, icddr,b and NGO health facilities. Of the women who visited a public health facility for their first ANC, 56%, 42%, and 26%, used public health facilities for their second, third and fourth ANC and 17% for childbirth. In contrast, 51% of them attended private health facilities for delivery.

Of the women (n = 1445) who received their first ANC in public facilities, only 6.3% attended ANC in public health facilities all four times. Among women (n = 1315) who received

**Table 1. Associations of timely first ANC visit, timely four ANC visits, and facility delivery with socio-demographic determinants: Unadjusted odds ratios from logistic regression analysis.**

| Characteristics (n = 3293) | N (%) | Timely first ANC visit (n = 3242) | | | Timely four ANC visits (n = 3242) | | | Facility delivery (n = 3293) | | |
|---|---|---|---|---|---|---|---|---|---|---|
| | | n (%) | Unadjusted OR (95% CI) | P-value | n (%) | Unadjusted OR (95% CI) | P-value | n (%) | Unadjusted OR (95% CI) | P-value |
| Age in years | | | | | | | | | | |
| <20 | 581 (18) | 339 (59) | 1 | | 20 (5) | 1 | | 437 (75) | 1 | |
| 20–30 | 2470 (75) | 1433 (59) | 1.00 (0.83–1.21) | 0.976 | 103 (6) | 1.23 (0.75–2.00) | 0.407 | 1826 (74) | 0.93 (0.76–1.15) | 0.523 |
| >30 | 242 (7) | 148 (63) | 1.20 (0.88–1.65) | 0.247 | 13 (7) | 1.63 (0.80–3.34) | 0.178 | 183 (76) | 1.02 (0.72–1.45) | 0.902 |
| Parity | | | | | | | | | | |
| 0 | 1306 (40) | 804 (62) | 1.25 (1.05–1.49) | 0.014 | 47 (4) | 0.69 (0.45–1.06) | 0.093 | 1014 (78) | 1.64 (1.35–1.99) | <0.001 |
| 1 | 1145 (35) | 653 (58) | 1.05 (0.87–1.26) | 0.607 | 47 (4) | 1.80 (0.52–1.23) | 0.309 | 860 (75) | 1.42 (1.17–1.73) | <0.001 |
| ≥ 2 | 842 (26) | 463 (57) | 1 | | 42 (5) | 1 | | 572 (68) | 1 | |
| Years of education | | | | | | | | | | |
| 0–5 | 426 (13) | 230 (56) | 1 | | 12 (4) | 1 | | 260 (61) | 1 | |
| 6–10 | 2285 (69) | 1302 (58) | 1.10 (0.89–1.36) | 0.366 | 98 (6) | 1.53 (0.83–2.81) | 0.173 | 1693 (74) | 1.83 (1.47–2.27) | <0.001 |
| >10 | 582 (18) | 388 (67) | 1.61 (1.24–2.08) | <0.001 | 26 (6) | 1.57 (0.78–3.15) | 0.204 | 493 (85) | 3.54 (2.62–4.76) | <0.001 |
| Husband education | | | | | | | | | | |
| 0–5 | 888 (27) | 464 (54) | 1 | | 36 (6) | 1 | | 577 (65) | 1 | |
| 6–10 | 1879 (57) | 1106 (60) | 1.26 (1.07–1.48) | 0.005 | 69 (5) | 0.88 (0.59–1.34) | 0.560 | 1433 (76) | 1.73 (1.46–2.06) | <0.001 |
| >10 | 526 (16) | 350(67) | 1.75 (1.39–2.19) | <0.001 | 31 (8) | 1.45 (0.88–2.37) | 0.141 | 436 (83) | 2.61 (2.00–3.41) | <0.001 |
| Wealth Index | | | | | | | | | | |
| Poorest | 671 (20) | 336 (52) | 1 | | 18 (4) | 1 | | 422 (63) | 1 | |
| Poorer | 669 (20) | 372 (57) | 1.22 (0.98–1.52) | 0.069 | 25 (6) | 1.39 (0.75–2.58) | 0.293 | 467 (70) | 1.36 (1.09–1.71) | 0.007 |
| Middle | 746 (23) | 463 (62) | 1.56 (1.26–1.93) | <0.001 | 35 (7) | 1.74 (0.98–3.10) | 0.060 | 585 (78) | 2.14 (1.70–2.71) | <0.001 |
| Richer | 552 (17) | 329 (60) | 1.42 (1.13–1.78) | 0.003 | 27 (7) | 1.83 (0.99–3.35) | 0.052 | 428 (78) | 2.04 (1.58–2.63) | <0.001 |
| Richest | 655 (20) | 420 (65) | 1.71 (1.37–2.14) | <0.001 | 31 (6) | 1.76(0.98–3.18) | 0.061 | 544 (83) | 2.89 (2.24–3.74) | <0.001 |

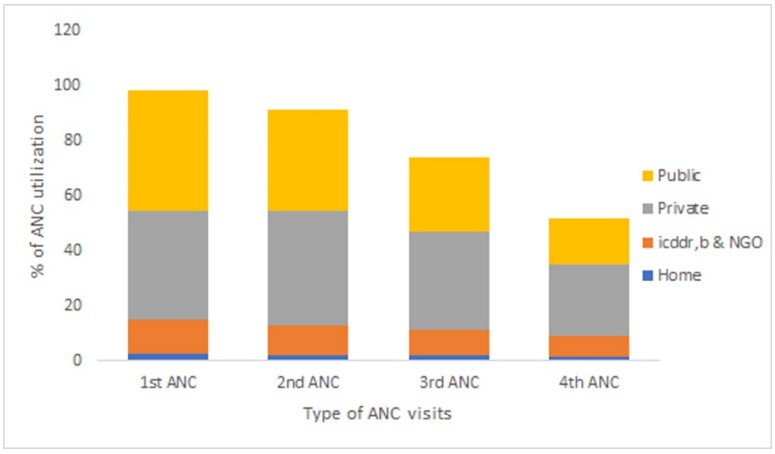

**Fig 1. Place of ANC utilization among all women.**

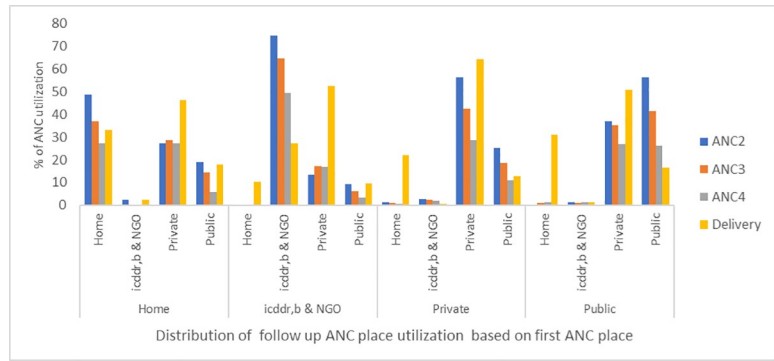

**Fig 2. Patterns of health facility use for subsequent antenatal and delivery care based on the place of the first antenatal care visit.**

their first ANC in private facilities, 5.2% received the ANC four times in private facilities. In the case of women (n = 398) who received their first ANC from icddr,b and NGOs, 3.8% of women went to NGO facilities for all four ANC.

Most women received their first ANC from doctors (Fig 3). A similar trend was seen for the type of healthcare provider for all four ANC visits (Fig 3). Doctors who provided the first ANC were mostly (92%) from private facilities, while 89% of nurses and midwives were from NGOs and icddr,b.

## Facility delivery

Among all respondents (n = 3293), 74% of women delivered in a health facility. Of the women who gave birth in a facility, 75% delivered in private facilities, 19% in public health facilities, and 6% in icddr,b, and NGO-led facilities. More than half of the deliveries (55%) were conducted by doctors (Fig 4); 92% of the doctors were from private facilities. About 80% of women used skilled birth attendants (SBA) during childbirth. Of the women who delivered their baby by normal vaginal birth, 41% of them were conducted by a Traditional Birth Attendant (TBA). About half of the participants (51%) delivered their baby by caesarian section and 92% of the cesarean sections occurred in private facilities.

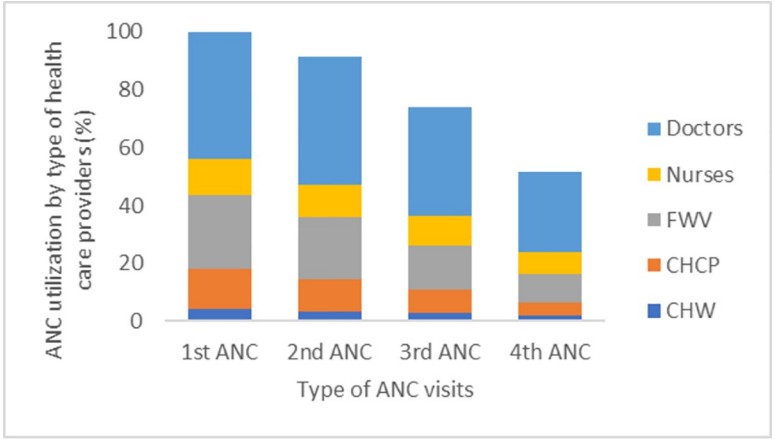

**Fig 3. ANC utilization by health care providers.** CHW (Community Health Workers: FWA, HA, others; FWV (Family Welfare Visitor); CHCP (Community Health Care Provider).

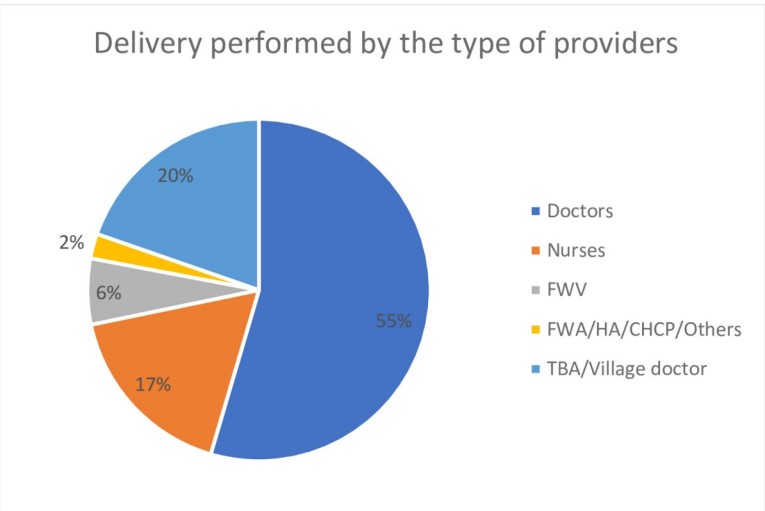

**Fig 4. Type of providers performing delivery.**

## Socio-economic characteristics, and utilization of ANC and facility delivery

For the first ANC visit (Fig 5), 55% of women from the poorest group and 36% from the richest group used public facilities, whereas 32% of the poorest group and 49% of the richest group used private health facilities. In icddr,b, and NGO-led facilities, utilization was 10.5% and 13% among the poorest and richest groups, respectively. For the 4th ANC visit and delivery, private health facilities were preferred by all socio-economic groups (Figs 5 and 6).

## Socio-demographic determinants of a timely first ANC visit, four timely ANC visits and facility delivery

Women's age, parity, husband's education, and socio-economic status were associated with a timely first ANC visit (Table 1). Women over 30 years of age were 1.5 times more likely than those less than 20 to attend their first ANC on time. Nulliparous women were 1.3 times more prone to attend timely for their first ANC compared to women with two or more births. Timely first ANC was 1.4 times more likely among the women whose husbands had completed more than ten years of education than if their husbands had 0–5 years of schooling. Women with higher socio-economic status were more likely to have a timely first ANC. We did not

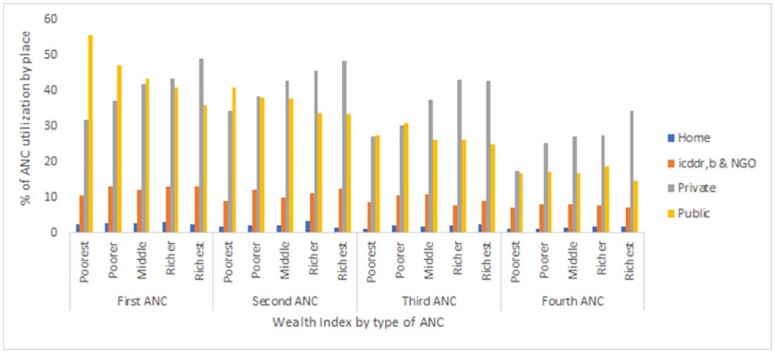

**Fig 5. Health facility utilization for ANC visits by wealth index.**

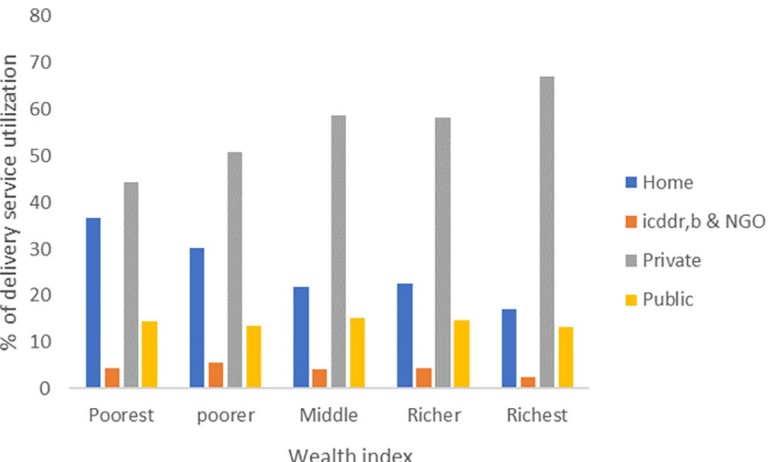

**Fig 6. Health facility utilization for facility delivery by wealth index.**

find any associations between the socio-demographic determinants used in our analysis and four timely ANC visits (Table 2).

Women's age, parity, education, husbands' education, and socio-economic status were associated with facility deliveries (Table 2). Women with 30 years of age or more were 1.6 times more prone to deliver in the health facilities than those less than 20. Nulliparous women were 1.5 times more likely to access a health facility for childbirth compared to women with two or more births. Women who had completed more than 10 years of school were 2 times more likely to deliver in a health facility in comparison to women with less than 5 years of education. Women whose husband had completed more than 10 years of school were 1.5 times more likely to deliver in a health facility than women whose husbands had less than 5 years of education. Women belonging to the richest wealth index category had 2.2 times higher odds to deliver in the health facility. Results with unadjusted odds ratios are presented in Table 1. Similar associations were found for skilled birth attendance as for facility delivery (S1 Table).

## Discussion

We found that six in ten women attended their first ANC timely, but that timely utilization of ANC four times as recommended was very low. Seven in ten women delivered in a health facility. Our study also found that women gradually moved from public health facilities to private hospitals, as their pregnancies advanced. The study identified socio-economic inequities in public versus private health facilities for utilization of ANC and childbirth care. We identified age, parity, wealth index, and education of the women's husbands as determinants of timely first ANC and facility delivery.

Utilization of private hospitals for maternal healthcare services is increasing all over Bangladesh. The unavailability of essential maternal healthcare supplies, services, and care providers in the public sector, as well as doctors practicing in both public and private sectors simultaneously, may encourage women to resort to more expensive private sector healthcare [28]. Similar trends were observed in other studies from low- and- middle income countries [29,30]. Women's negative perceptions and experiences of public health care could also drive them to private health facilities and away from public health facilities [29,31]. Women's and family members' preference to receive care from doctors, especially during the last trimester of pregnancy and for delivery, may drive care-seeking toward private health facilities. Our study confirms that, of the women receiving care from doctors, the majority were from private

**Table 2. Associations of timely first ANC visit, timely four ANC visits, and facility delivery with socio-demographic determinants: Adjusted odds ratios from logistic regression analysis.**

| Characteristics (n = 3293) | N (%) | Timely first ANC visit (n = 3242) | | | Timely four ANC visits (n = 3242) | | | Facility delivery (n = 3293) | | |
|---|---|---|---|---|---|---|---|---|---|---|
| | | n (%) | Adjusted OR* (95% CI) | P-value | n (%) | Adjusted OR* (95% CI) | P-value | n (%) | Adjusted OR* (95% CI) | P-value |
| **Age in years** | | | | | | | | | | |
| <20 | 581 (18) | 339 (59) | 1 | | 20 (5) | 1 | | 437 (75) | 1 | |
| 20–30 | 2470 (75) | 1433 (59) | 1.16 (0.93–1.45) | 0.202 | 103 (6) | 1.07 (0.60–1.93) | 0.803 | 1826 (74) | 1.14 (0.88–1.48) | 0.317 |
| >30 | 242 (7) | 148 (63) | 1.52 (1.05–2.19) | 0.026 | 13 (7) | 1.21 (0.52–2.86) | 0.657 | 183 (76) | 1.60 (1.05–2.43) | 0.028 |
| **Parity** | | | | | | | | | | |
| 0 | 1306 (40) | 804 (62) | 1.30 (1.04–1.62) | 0.022 | 47 (4) | 0.68 (0.40–1.16) | 0.152 | 1014 (78) | 1.52 (1.18–1.95) | 0.001 |
| 1 | 1145 (35) | 653 (58) | 1.06 (0.88–1.29) | 0.530 | 47 (4) | 0.79 (0.50–1.25) | 0.324 | 860 (75) | 1.38 (1.12–1.71) | 0.003 |
| ≥ 2 | 842 (26) | 463 (57) | 1 | | 42 (5) | 1 | | 572 (68) | 1 | |
| **Years of education** | | | | | | | | | | |
| 0–5 | 426 (13) | 230 (56) | 1 | | 12 (4) | 1 | | 260 (61) | 1 | |
| 6–10 | 2285 (69) | 1302 (58) | 0.94 (0.75–1.18) | 0.613 | 98 (6) | 1.57 (0.83–2.97) | 0.166 | 1693 (74) | 1.38 (1.09–1.74) | 0.007 |
| >10 | 582 (18) | 388 (67) | 1.13 (0.84–1.52) | 0.422 | 26 (6) | 1.46 (0.66–3.21) | 0.349 | 493 (85) | 1.97 (1.40–2.77) | <0.001 |
| **Husband education** | | | | | | | | | | |
| 0–5 | 888 (27) | 464 (54) | 1 | | 36 (6) | 1 | | 577 (65) | 1 | |
| 6–10 | 1879 (57) | 1106 (60) | 1.14 (0.96–1.36) | 0.139 | 69 (5) | 0.78 (0.51–1.21) | 0.264 | 1433 (76) | 1.31 (1.09–1.58) | 0.005 |
| >10 | 526 (16) | 350(67) | 1.40 (1.09–1.81) | 0.009 | 31 (8) | 1.25 (0.71–2.19) | 0.435 | 436 (83) | 1.53 (1.14–2.06) | 0.005 |
| **Wealth Index** | | | | | | | | | | |
| Poorest | 671 (20) | 336 (52) | 1 | | 18 (4) | 1 | | 422 (63) | 1 | |
| Poorer | 669 (20) | 372 (57) | 1.17 (0.94–1.46) | 0.164 | 25 (6) | 1.40 (0.75–2.60) | 0.291 | 467 (70) | 1.23 (0.98–1.56) | 0.079 |
| Middle | 746 (23) | 463 (62) | 1.45 (1.17–1.81) | 0.001 | 35 (7) | 1.71 (0.95–3.08) | 0.076 | 585 (78) | 1.84 (1.44–2.34) | <0.001 |
| Richer | 552 (17) | 329 (60) | 1.27 (1.00–1.62) | 0.046 | 27 (7) | 1.74 (0.93–3.26) | 0.083 | 428 (78) | 1.64 (1.26–2.14) | <0.001 |
| Richest | 655 (20) | 420 (65) | 1.49 (1.18–1.89) | 0.001 | 31 (6) | 1.67 (0.89–3.11) | 0.109 | 544 (83) | 2.15 (1.63–2.82) | <0.001 |

*Adjusted with women's age, parity, education, husband's education, and wealth index.

hospitals. Our study findings are also align with the report of the Bangladesh Demographic and Health Survey [14]. A study conducted in Bangladesh reported increased utilization of private health facilities compared to public health facilities for maternal health care services over time [32]. Basic and comprehensive packages of emergency obstetric care with skilled providers, especially doctors, may need to be routinely available throughout the public sector to increase retention of women who start their ANC in hopes that they remain there for delivery care.

Across LMIC in Asia and Africa, there are wide within- and between-country variations in coverage of timely ANC initiation [33–39]. The coverage of first ANC within 16 weeks from a skilled provider in our study is higher (53%) than reported in a study (32%) conducted in three northern districts of Bangladesh [25]. Our study also shows somewhat higher coverage of timely first ANC compared to the national coverage reported in the most recent Demographic and Health Survey (37%) [14]. Another study conducted in Bangladesh measuring initiation of ANC, as a first visit within 12 weeks of gestation, also found low coverage [40]. Implementation of the Demand-side Financing Maternal Health Voucher Scheme in this area

by the MOHFW in Bangladesh since 2010, might explain the higher coverage in our study compared to national reports [41].

Although half of women received ANC four or more times, our study found low coverage of four timely ANC visits illustrating that women might attend an adequate number of times, but they lacked actionable information on the nationally recommended ANC visits schedule. The coverage of four timely ANC visits was found to be low despite being relatively high for first timely ANC. A possible explanation could be women visited the public health facility for their first ANC as the health facility was very close to their house. However, by the time of the fourth visit, they had experienced the lack of available services and may have tried to reach additional essential services such as ultrasonogram which is primarily available in the private facilities. Other studies conducted in northern Bangladesh and Ethiopia found low coverage of four timely ANC visits despite high uptake of a timely first ANC [25,42]. In Ethiopia, though their first ANC coverage is less than ours, their timely four ANC utilization was notably greater than this study [42]. This illustrates that the drivers of timely first ANC and timely four ANC visits may differ both within and between countries. More information is needed to understand these differences. We need to assess the context-specific mechanisms to increase timely ANC utilization. Reminders by phone call or text message could be employed to increase the timeliness of ANC attendance [43–45].

Our results provide additional supportive evidence that institutional delivery (74%) is still increasing, including in this particular area. A survey conducted in 2015 in our study area reported a coverage of 49% [46]. The higher coverage in our results may be attributed to the implementation of several interventions to increase facility delivery in the study area, including strengthened health education activities by health workers [47]. Demand-side Financing Maternal Health Voucher Scheme for ultra-poor women is another such intervention [41,48]. Other reasons might include increased availability of and access to health facilities as well as overall improvements in the population's economic status [49,50].

The women's husbands' education appears to be strongly associated with early initiation of ANC and facility delivery. In Bangladesh, men often have the privilege of making decisions for their wives, which may explain the association between the husbands' education and maternity service utilization [19,20,39,51]. Our results also show that more educated women were more likely to have an institutional delivery. This may be attributed to the fact that women with higher education have a better understanding of seeking care, more awareness of the value of health care utilization, and making decisions with confidence [19,20,52].

This study also found that women's economic status was a strong predictor for compliance with the nationally recommended first ANC contact and facility delivery, with the richest women more likely to seek early health care. Our results are similar to previous studies that found a positive association between the economic status of women and early initiation of ANC and facility delivery, probably due to ease of access to health care [50,53,54]. This study also identified disparities in the use of public and private health facilities for ANC and childbirth care services in Bangladesh's rural context. Poor women used public health facilities more often, while rich women were more likely to use private facilities. Designing interventions addressing inequities in maternal health service utilization is important to increase access in those with the highest need for support.

The study was conducted to assess patterns of health facility utilization in Matlab. Our evidence suggested that women, irrespective of rich and poor, typically shift frequently between the public and private health systems, possibly to access better quality of care. Continuity of care helps improve the quality of care by establishing good relationships and trust between the provider and the woman [55,56]. A comprehensive description of women's transition between facilities and types of facilities is important to understand women's decision-making processes

and reasoning and is necessary to design appropriate health care interventions to improve care seeking, continuity and timeliness.

## Strengths and limitations

This study has several strengths. Our study is a large, cross-sectional population-based study. Women were interviewed within a short period after birth potentially minimizing recall bias. Most studies of maternal healthcare utilization and coverage have used survey data, where women with a live birth in the two-three years before the survey were interviewed, which might introduce recall bias. We used a standardized questionnaire to ensure comparable responses from the participants. We used an asset-based index, which is a good proxy for measuring household wealth status in this community. One of the limitations of the study lies in the accurate assessment of gestational age at the ANC visit. This was available for a subset of women through the eRegistry, but we chose to use the data from recall for consistency. Women's recall of the dates of ANC visits could be incomplete or faulty when such data are collected retrospectively through household surveys [57]. Unmeasured predisposing, enabling, and need-based factors not included in our analysis might affect coverage of timely first ANC, timely four ANC visits, and facility delivery. Coverage of four timely visits was low, which precluded meaningful analysis of associations.

## Conclusions and recommendations

To increase the health service utilization and promote good health, maternal health care programs should pay particular attention to young, multiparous women, of low socio-economic status, or with poorly educated husbands. The reasons pregnant women, irrespective of wealth shifted from public to private facilities need to be explored through mixed methods research to inform policy makers, planners and program designers to intervene in the most appropriate areas. Further research is required to understand which factors are the true drivers of timely health care utilization and how to translate these drivers into improved policies and interventions which ultimately strengthen timely maternity care utilization.

## Supporting information

**S1 Table. Associations of skilled birth attendance at delivery with socio-demographic determinants.**
(DOCX)

**S1 Dataset.**
(DTA)

**S1 File. Questionnaire.**
(PDF)

## Acknowledgments

We would like to forward our gratitude to all the respondents who participated in this study.

## Author Contributions

**Conceptualization:** Jesmin Pervin, Mahima Venkateswaran, Ingrid K. Friberg, Anisur Rahman, J. Frederik Frøen.

**Data curation:** Jesmin Pervin, U. Tin Nu, Monjur Rahman.

**Formal analysis:** Jesmin Pervin, Mahima Venkateswaran, U. Tin Nu, Monjur Rahman, Brian F. O'Donnell, Ingrid K. Friberg, Anisur Rahman, J. Frederik Frøen.

**Funding acquisition:** J. Frederik Frøen.

**Methodology:** Jesmin Pervin, Anisur Rahman, J. Frederik Frøen.

**Project administration:** Jesmin Pervin, U. Tin Nu, Monjur Rahman, Ingrid K. Friberg, Anisur Rahman.

**Supervision:** Jesmin Pervin, Ingrid K. Friberg, Anisur Rahman.

**Writing – original draft:** Jesmin Pervin.

**Writing – review & editing:** Jesmin Pervin, Mahima Venkateswaran, U. Tin Nu, Monjur Rahman, Brian F. O'Donnell, Ingrid K. Friberg, Anisur Rahman, J. Frederik Frøen.

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
