## [Decision Letter · Decision Letter 0]

23 Jul 2021

PONE-D-21-17032

Determinants of utilization of antenatal and delivery care at the community level in rural Bangladesh

PLOS ONE

Dear Dr. Pervin,

Thank you for submitting your manuscript to PLOS ONE. After careful consideration, we feel that it has merit but does not fully meet PLOS ONE’s publication criteria as it currently stands. Therefore, we invite you to submit a revised version of the manuscript that addresses the points raised during the review process.

We look forward to receiving your revised manuscript.

Kind regards,

Russell Kabir, PhD

Academic Editor

PLOS ONE

Journal Requirements:

2. Please include additional information regarding the survey or questionnaire used in the study and ensure that you have provided sufficient details that others could replicate the analyses. For instance, if you developed a questionnaire as part of this study and it is not under a copyright more restrictive than CC-BY, please include a copy, in both the original language and English, as Supporting Information. If the original language is written in non-Latin characters, for example Amharic, Chinese, or Korean, please use a file format that ensures these characters are visible.

3. Please state whether you validated the questionnaire prior to testing on study participants. Please provide details regarding the validation group within the methods section.

4. Thank you for stating the following in the Funding Section of your manuscript:

“This research is part of the eRegistries Bangladesh project funded by the Norwegian Research Council (grant agreement number 248073/H10; title: Strengthening the extension of Reproductive, Maternal, Newborn, and Child Health services in Bangladesh with an electronic health registry: A cluster randomized controlled trial), and the Centre for Intervention Science in Maternal and Child Health (CISMAC), Center for International Health, University of Bergen (project number: 223269). icddr,b is also grateful to the Governments of Bangladesh, Canada, Sweden and the UK for providing core/unrestricted support. The funders had no role in the design of the study and collection, analysis, and interpretation of data.”

 “This research is part of the eRegistries Bangladesh project funded by the Norwegian Research Council (grant agreement number 248073/H10; title: Strengthening the extension of Reproductive, Maternal, Newborn, and Child Health services in Bangladesh with an electronic health registry: A cluster randomized controlled trial), and the Centre for Intervention Science in Maternal and Child Health (CISMAC), Center for International Health, University of Bergen (project number: 223269). icddr,b is also grateful to the Governments of Bangladesh, Canada, Sweden and the UK for providing core/unrestricted support. The funders had no role in the design of the study and collection, analysis, and interpretation of data.”

“No authors have competing interests.”

Reviewers' comments:

Reviewer's Responses to Questions

**Comments to the Author**

1. Is the manuscript technically sound, and do the data support the conclusions?

Reviewer #1: Yes

Reviewer #2: Yes

Reviewer #3: Yes

2. Has the statistical analysis been performed appropriately and rigorously? 

Reviewer #1: Yes

Reviewer #2: No

Reviewer #3: Yes

3. Have the authors made all data underlying the findings in their manuscript fully available?

Reviewer #1: No

Reviewer #2: Yes

Reviewer #3: Yes

4. Is the manuscript presented in an intelligible fashion and written in standard English?

Reviewer #1: Yes

Reviewer #2: Yes

Reviewer #3: Yes

5. Review Comments to the Author

Reviewer #1: The questions were well thought and appropriate and literature review has been adequate as well. However although author did mention about the shifting of all pregnant women from public to private hospital for delivery but did not mention about the cause and also did not backed up by adequate data. The determinants did mention about the economic dimension which is crucial and there should be significant financial barrier of the people below poverty line. The findings and recommendation could have been further improved to come up with few recommendations related to shifting. However, the author tried to focus on the determinants and limited the analysis which may need an in-depth study to understand the dynamics in future research. The figures and tables have been clean and readable and so far accurate. The figures and tables though supports the findings yet the author could also need to adopt qualitative method to learn about the reasons why the people irrespective of poor and rich did shift finally to the private hospital as we all know this could lead someone fall into poverty trap. Since the study was conducted in rural Bangladesh, the authors could also add few questions and further enrich the study to inform the policy makers, planners and program people to intervene in that areas. However I would recommend a follow up study could be undertaken using mixed method if possible to come up with few recommendations. The study used appropriate method and the conclusion has been drawn based on the findings or the results and discussions are all supportive to the conclusion of the study. The author did mention about the limitation of study. The study also has the scope to be validated by others or can recreated for further analysis. The study findings and conclusion are found fully aligned with the claims of the author.

Reviewer #2: The manuscript prepared by Pervin and colleagues is quite a common work in Bangladesh. Several previous studies nearly same content have been published, but not in the same location. Study gap is deeply missed here which is the mandatory segment of a good article. Statistical analysis seems unclear to me. Significance level of each step of analysis is highly recommended.

#The introduction misses the global statistics of ANC specially the neighboring countries of Bangladesh. It mostly presents the national data. So authors should try to store more information regarding this.

#Why these two-subdivisions, why not the whole district Chandpur? What do the places signify for this work?

#Authors are suggested to modify the result graphs (pie, line, bar etc.). All are made with same design and color.

#Table S1 needs highlighting the heading points and the name of variables.

#Result tables are not organized. Authors represented a single final table without p-value. I am not satisfied at this point. Either modify with separate univariable and multivariable table or provide valid explanation.

#Describe the source of categorizing the wealth index.

#Linguistic improvement is needed in introduction and discussion part.

Reviewer #3: Title- title is clear, concise, informative.

Abstract- Abstract is included, outline methodology, provided sample subjects, reported major findings. In the background section could have provided research problem.

Introduction - Problem clearly identified and rationale of the study stated.

Literature review - literature review is up-to-date and presented a balanced evaluation.

Methodology - methodology clearly stated, subjects clearly identified, sample selection and sample size stated, data collection procedures adequately described, validity and reliability of the questionnaire clearly stated. methodology section is the strength of the study.

Results - results are clear, internally consistent, sufficient detail is given to enable reader to have confidence on findings, tables and graphs have been provided to present the results.

Data Analysis - statistical analysis performed correctly, complete information is provided.

Discussion - discussion draws upon previous researches, strengths and weaknesses are provided. Could have compared with few recent ANC studies to make discussion balanced.

Conclusion and Recommendations - these two sections has been covered under discussion. conclusion has been supported by results, recommendations suggest further areas for research. Could have been added separate section for conclusion.

Comments :

1. In background section of abstract, research problem has not been mentioned. Just the rationale of the study provided. Could authors please add the research problem identified.

2. In the discussion could authors please compare these three research papers listed below:

Kabir, R., Majumder, A.A., Arafat, S.Y., Chodwhury, R.K., Sultana, S., Ahmed, S.M., Monte-Serrat, D.M. and Chowdhury, E.Z., 2018. Impact of Intimate Partner violence on ever married women and utilization of antenatal care services in Tanzania. Journal of College of Medical Sciences-Nepal, 14(1), pp.7-13.

Kabir, R. and Khan, H., 2013. Utilization of Antenatal care among pregnant women of Urban Slums of Dhaka City, Bangladesh. IOSR Journal of Nursing and Health Science, 2(2).

Kabir, R., Haider, M.R. and Kordowicz, M., 2018. A Cross-sectional study to explore the challenges faced by Myanmar women in accessing antenatal care services. Epidemiology Biostatistics and Public Health, 15(3), p.e12933.

3. Authors have concluded the study and provided the recommendations but there is no separate section for it. It would be good if there is a specific section for conclusion and recommendation to make it easy for the readers.

6. PLOS authors have the option to publish the peer review history of their article (what does this mean?). If published, this will include your full peer review and any attached files.

Reviewer #1: **Yes: **Munir Ahmed

Reviewer #2: No

Reviewer #3: **Yes: **Divya Vinnakota

---

## [Author Response · Author response to Decision Letter 0]

1 Sep 2021

Review Response PONE-D-21-17032

Determinants of utilization of antenatal and delivery care at the community level in rural Bangladesh

PLOS ONE

Thanks for giving us the opportunity to revise our manuscript. Please find below the comments made by editor’s and reviewers followed by our responses.

Editors comments:

Response: Thanks, ensured.

2. Please include additional information regarding the survey or questionnaire used in the study and ensure that you have provided sufficient details that others could replicate the analyses. For instance, if you developed a questionnaire as part of this study and it is not under a copyright more restrictive than CC-BY, please include a copy, in both the original language and English, as Supporting Information. If the original language is written in non-Latin characters, for example Amharic, Chinese, or Korean, please use a file format that ensures these characters are visible.

Response: We have included English and Bangla questionnaire as suggested (S1 questionnaire).

3. Please state whether you validated the questionnaire prior to testing on study participants. Please provide details regarding the validation group within the methods section.

Response: We used the questionnaire from the previous study conducted in the same area where the sample size was 2262 (Pervin et al., 2018). We used shorter version of the questionnaire based on the need of our study. Data collectors were trained for data collection and the questionnaire was pretested on 40 pregnant women. After the pretest, a discussion was carried out on responses with all of the interviewer to ensure the uniformity of the potential responses. 

 4. Thank you for stating the following in the Funding Section of your manuscript:

“This research is part of the eRegistries Bangladesh project funded by the Norwegian Research Council (grant agreement number 248073/H10; title: Strengthening the extension of Reproductive, Maternal, Newborn, and Child Health services in Bangladesh with an electronic health registry: A cluster randomized controlled trial), and the Centre for Intervention Science in Maternal and Child Health (CISMAC), Center for International Health, University of Bergen (project number: 223269). icddr,b is also grateful to the Governments of Bangladesh, Canada, Sweden and the UK for providing core/unrestricted support. The funders had no role in the design of the study and collection, analysis, and interpretation of data.”

 “This research is part of the eRegistries Bangladesh project funded by the Norwegian Research Council (grant agreement number 248073/H10; title: Strengthening the extension of Reproductive, Maternal, Newborn, and Child Health services in Bangladesh with an electronic health registry: A cluster randomized controlled trial), and the Centre for Intervention Science in Maternal and Child Health (CISMAC), Center for International Health, University of Bergen (project number: 223269). icddr,b is also grateful to the Governments of Bangladesh, Canada, Sweden and the UK for providing core/unrestricted support. The funders had no role in the design of the study and collection, analysis, and interpretation of data.” 

Response: We have removed funding-related text from the manuscript and added statements in the cover letter.

“No authors have competing interests.”

Response: We have included the information in the cover letter.

Response: Sorry for the confusion. We wanted to mean that data was not shown in the manuscript. We revised the text in the manuscript (Line no:296, page no:14).

Response: We have reviewed the references and didn’t include any retracted paper.

Reviewers' comments:

Reviewer #1

The questions were well thought and appropriate and literature review has been adequate as well. However although author did mention about the shifting of all pregnant women from public to private hospital for delivery but did not mention about the cause and also did not backed up by adequate data. The determinants did mention about the economic dimension which is crucial and there should be significant financial barrier of the people below poverty line. The findings and recommendation could have been further improved to come up with few recommendations related to shifting. However, the author tried to focus on the determinants and limited the analysis which may need an in-depth study to understand the dynamics in future research. The figures and tables have been clean and readable and so far accurate. The figures and tables though supports the findings yet the author could also need to adopt qualitative method to learn about the reasons why the people irrespective of poor and rich did shift finally to the private hospital as we all know this could lead someone fall into poverty trap. Since the study was conducted in rural Bangladesh, the authors could also add few questions and further enrich the study to inform the policy makers, planners and program people to intervene in that areas. However I would recommend a follow up study could be undertaken using mixed method if possible to come up with few recommendations. The study used appropriate method and the conclusion has been drawn based on the findings or the results and discussions are all supportive to the conclusion of the study. The author did mention about the limitation of study. The study also has the scope to be validated by others or can recreated for further analysis. The study findings and conclusion are found fully aligned with the claims of the author.

Response: Thank you for your thoughtful insights. We have already recommended for the future research in the discussion section. We believe that future research through mixed method approach is required to find in-depth insights into the reason for shifting pregnant women irrespective of rich and poor from public to private hospitals to inform the policy makers, planners and program people to intervene in that areas. We have added the statement in the recommendation section (line: 395-403, page: 21).

Reviewer 2: 

1. The manuscript prepared by Pervin and colleagues is quite a common work in Bangladesh. Several previous studies nearly same content have been published, but not in the same location. Study gap is deeply missed here which is the mandatory segment of a good article. Statistical analysis seems unclear to me. Significance level of each step of analysis is highly recommended.

Response: Thank you for your observation. We have revised the research gap in the introduction section and data analysis in the method section to make clearer (line: 87-93, 186-212 page: 5, 9-10).

2. The introduction misses the global statistics of ANC specially the neighboring countries of Bangladesh. It mostly presents the national data. So authors should try to store more information regarding this.

Response: We have already mentioned the global statistics of ANC in the introduction section, In the revised version we have added south Asian statistics (line: 60 - 61, page: 3).

3. Why these two-subdivisions, why not the whole district Chandpur? What do the places signify for this work?

Response: This study was on add-on to the specific study titled “eRegMat” that was conducted in two sub-districts, Matlab South and Matlab North under Chandpur district of Bangladesh. Therefore, we were limited by the population size. Furthermore, the “eRegMat” was a cluster randomized trial and the facilities within the two sub-districts were adequate in number for the required sample size. 

4. Authors are suggested to modify the result graphs (pie, line, bar etc.). All are made with same design and color.

Response: Thank you for your suggestion. One of our bar graphs revised into pie graph (Fig 4). We have been uniformed with our color scheme and we therefore defer to the layout and color decision of the graphs made by the journal. 

5. Table S1 needs highlighting the heading points and the name of variables.

Response: The heading points and the name of variables highlighted in Table1, Table2 and S1 Table (line:298-299, 300-301, 563, page:15-16,25).

6. Result tables are not organized. Authors represented a single final table without p-value. I am not satisfied at this point. Either modify with separate univariable and multivariable table or provide valid explanation.

Response: We have moved the univariate (unadjusted) analysis in the main manuscript as table 1 from appendix to be presented before adjusted results. We revised unadjusted and adjusted result table (Table 2) with odds ratio, 95% CI and p-value in the manuscript (line: 298-301 page: 15-16).

7. Describe the source of categorizing the wealth index.

 Response: Reference added as the source of categorizing the wealth index (line:171-172, page:8-9).

8. Linguistic improvement is needed in the introduction and discussion part.

Response: The full manuscript was reviewed by Native English authors.

Reviewer 3

Title- title is clear, concise, informative.

Abstract- Abstract is included, outline methodology, provided sample subjects, reported major findings. In the background section could have provided research problem.

Introduction - Problem clearly identified and rationale of the study stated.

Literature review - literature review is up-to-date and presented a balanced evaluation.

Methodology - methodology clearly stated, subjects clearly identified, sample selection and sample size stated, data collection procedures adequately described, validity and reliability of the questionnaire clearly stated. methodology section is the strength of the study.

Results - results are clear, internally consistent, sufficient detail is given to enable reader to have confidence on findings, tables and graphs have been provided to present the results.

Data Analysis - statistical analysis performed correctly, complete information is provided.

Discussion - discussion draws upon previous researches, strengths and weaknesses are provided. Could have compared with few recent ANC studies to make discussion balanced.

Conclusion and Recommendations - these two sections has been covered under discussion. conclusion has been supported by results, recommendations suggest further areas for research. Could have been added separate section for conclusion.

Response: Thank you for your beneficial feedback.

Comments:

1. In background section of abstract, research problem has not been mentioned. Just the rationale of the study provided. Could authors please add the research problem identified.

Response: Research problem added in the background section of the abstract (line:21-22, page:2).

2. In the discussion could authors please compare these three research papers listed below:

i. Kabir, R., Majumder, A.A., Arafat, S.Y., Chodwhury, R.K., Sultana, S., Ahmed, S.M., Monte-Serrat, D.M. and Chowdhury, E.Z., 2018. Impact of Intimate Partner violence on ever married women and utilization of antenatal care services in Tanzania. Journal of College of Medical Sciences-Nepal, 14(1), pp.7-13.

ii. Kabir, R. and Khan, H., 2013. Utilization of Antenatal care among pregnant women of Urban Slums of Dhaka City, Bangladesh. IOSR Journal of Nursing and Health Science, 2(2).

iii. Kabir, R., Haider, M.R. and Kordowicz, M., 2018. A Cross-sectional study to explore the challenges faced by Myanmar women in accessing antenatal care services. Epidemiology Biostatistics and Public Health, 15(3), p.e12933.

Response: Thank you for your thoughts. We have only discussed on the factors identified in Bangladesh. Therefore, we have added the study used urban health survey data, identified risk factors of ANC in the introduction section as reference (line: 85, page: 5). 

3. Authors have concluded the study and provided the recommendations but there is no separate section for it. It would be good if there is a specific section for conclusion and recommendation to make it easy for the readers.

Response: A separate section was made for conclusion and recommendations (line: 395-403, page: 21). 

Additional Reviewers’ comments in the manuscript:

Abstract

1. How did you determine the association? Mention the significance level/P-value in each step of analysis and justify.

Response: All the variables presented in the study were considered in a directed acyclic graph (DAG) approach and identified all the variable as potential confounder in the causal association. We also identified multicollinearity by correlation coefficient and variance inflation factor. We determined the association through univariate and multivariate logistic regression analysis. The associations of independent and explanatory variables in the multivariate logistic regression were considered as statistically significant if the p-value was found to be <0.05. The result tables were revised with odds ratio, 95% CI and p-value. We have revised the text in the method section of the abstract (line no: 27-31 Page: 2). 

Introduction

2. Timely utilization of antenatal care (ANC) and delivery services is important to support maternal health and allow for the best possible start to babies' lives. 

Response: Thank you for the observation. The full manuscript was reviewed by the English Native authors (line no: 50, page: 3).

3. Paraphrase the word “utilization” and rewrite sentence “Despite progress in maternal health service utilization in some low- and middle-income countries, further increase in utilization is needed to impact maternal and neonatal health”

Response: We have revised the sentence (line no: 55-57, page 3).

4. Place this paragraph just after first one. 

Response: We moved the paragraph accordingly (line no: 58-63, page 3-4).

5. Avoid writing multiple use of the word "utilization" 

Response: We have revised accordingly. 

6. What is the specific source of this information? Is this in reference '5'? Where? 

Response: We have added reference “14”. It is in the table 9.3 of page 133 (line: 74, page: 4).

7. “About half of all deliveries occur in health facilities, most of them privately-owned” 

Response: We have revised the sentence (line: 75, page: 4).

8. What is the research gap of this study? Last paragraph of introduction should clearly present the statement. 

Response: We have revised the paragraph (line: 87-93, page: 5).

9. The sentence starts alike the previous one. Alteration is suggested. 

Response: We have revised the sentence (line: 89-91, page: 5).

Method

10. This paragraph seems too long. Readers would be bored...It is not necessary to describe elaborately each strata of Govt. health sectors. You just describe sampling design briefly! 

Response: We have revised the paragraph accordingly (line: 101-115, page: 5-6).

11. What is the sample size of piloting study? 

Response: The questionnaire was pretested on 40 pregnant women. We have already mentioned about the pretesting in the editor’s response of comment no 3. 

12. Mention the formula which you used! 

Response: We used “power oneproportion” and “powerlog” command in the Stata (version 16) and added in the text of the sample size of the method section (line: 183-185, page: 9). 

13. What was your calculated sample size? 

Response: This study is a sub-study of the eRegMat trial. Therefore, we had a fixed sample size and we did this power calculation to assess adequacy for the intended outcome measures.

14. Describe stage by stage what was your model analysis. As like.. 

-univariable analysis (test name and p-value cut off)

-Multivariable model

-Correlation check (cut-off of R square)

-Goodness-of-test etc.

Response: We revised the data analysis in the method section (line no: 186 -212, page 9-10). 

15. Why only chi square test? Are all the explanatory variables categorical in the study? If thus, please mention the type of variables clearly. 

Response: All explanatory variables were categorized and we have revised in the method section (line: 186-196, page: 9-10).

16. Did you try with cut off 0.2 at univariable analysis step? If not, why? 

Response: In the method section we described analysis step by step. This was not the stepwise regression analysis (line: 186 -212, page: 9-10).

17. In the basis of what? mean, median or quantile? Please mention. 

Response: Parity was categorized based on median. Women and husbands’ education were categorized based on primary schooling (0-5yrs), secondary schooling (6-10 yrs) and higher secondary >10 yrs. The studies in Bangladesh used this standard category for the analysis.

Women age category was done based on the available studies and population group of this study.

18. At what extent of changes did you allow for not confounding? 

Response: We did not follow the stepwise regression analysis (line no: 186 -212, page: 9-10).

Results

19. How do you define this two groups? 

Response: We revised the groups in the method section and in the result section table (line no: 171-172, 268-295 page 8, 13-14).

20. Paraphrasing needed! Eg. 1.3 times higher/at risk/vulnerable/prone etc. Each underlines are suggested to be changed. 

Response: We have revised the sentence (line: 279-295, page: 14).

21. Is this the only result table? 

Where is the significance value/P-value of your variables?

Where is the univariable and multivariable analysis table?

How do you finalize the risk factors without examining the significance level of a character?

Where is the P-value????

Response: We moved the univariate analysis table from Appendix to the main manuscript. We have added p-values in the result table 1 and 2 (line no: 298-301, page 15-16) and we described step by step analysis in the method section (line: 186 -212, page: 9-10).

22. Table needs highlighting the heading points and the name of variables. 

Response: We have highlighted accordingly (line: 298-301, page: 15-16).

23. Better write 0.93 instead of .93 

 Likely for each...

 Response: Revised (line: 298-301, page: 15-16).

Discussion

24. What do you mean by dual practice? 

Response: We have revised the sentence to make clearer (line: 310-313 page: 17).

25. Confirms 

Response: Revised (line: 317-319, page: 17).

26. This finding totally agrees the report of DHS... 

Response: We revised the sentence (line: 319-320, page: 17).

27. Rewrite the sentence “They were not oriented then about the availability of services”

Response: We revised the sentence (line: 339-341, page: 18).

28. “Demand side financing for ultra-poor women”- whats that 

Response: We revised the sentence (line: 331-333, 353-354, page: 18- 19).

29. Too many citations! 2-3 are enough. 

Response: We have reduced the number of citations (line: 363, page: 19).

30. This paragraph should be continued with Discussion. Statements can be summarized under separate heading. 

Response: We kept strength and limitation under separate heading (line: 381-394, page: 20).

31. Naming the last part like "Conclusions and Recommendations" for better understanding of readers. 

Response: We made separate heading for last paragraph as conclusion and recommendation (line: 395-403, page: 21).

References:

Pervin, J., Nu, U. T., Rahman, A. M. Q., Rahman, M., Uddin, B., Razzaque, A., . . . Rahman, A. (2018). Level and determinants of birth preparedness and complication readiness among pregnant women: A cross sectional study in a rural area in Bangladesh. PLoS ONE, 13(12), e0209076. doi:10.1371/journal.pone.0209076

---

## [Editor Report · Decision Letter 1]

10 Sep 2021

Determinants of utilization of antenatal and delivery care at the community level in rural Bangladesh

PONE-D-21-17032R1

Dear Dr. Pervin,

We’re pleased to inform you that your manuscript has been judged scientifically suitable for publication and will be formally accepted for publication once it meets all outstanding technical requirements.

Kind regards,

Russell Kabir, PhD

Academic Editor

PLOS ONE
---

## [Editor Report · Acceptance letter]

20 Sep 2021

PONE-D-21-17032R1 

Determinants of utilization of antenatal and delivery care at the community level in rural Bangladesh 

Dear Dr. Pervin:

I'm pleased to inform you that your manuscript has been deemed suitable for publication in PLOS ONE. Congratulations! Your manuscript is now with our production department. 

Kind regards, 

on behalf of

Dr. Russell Kabir 

Academic Editor

PLOS ONE